# Natural-Product-Inspired Microwave-Assisted Synthesis of Novel Spirooxindoles as Antileishmanial Agents: Synthesis, Stereochemical Assignment, Bioevaluation, SAR, and Molecular Docking Studies

**DOI:** 10.3390/molecules28124817

**Published:** 2023-06-16

**Authors:** Nawal Kishore Sahu, Ritu Sharma, Kshirsagar Prasad Suhas, Jyoti Joshi, Kunal Prakash, Richa Sharma, Ramendra Pratap, Xiwen Hu, Sukhbir Kaur, Mukesh Jain, Carmine Coluccini, Paolo Coghi, Sandeep Chaudhary

**Affiliations:** 1Laboratory of Organic and Medicinal Chemistry (OMC Lab), Department of Chemistry, Malaviya National Institute of Technology, Jawaharlal Nehru Marg, Jaipur 302017, India; nawal.chemistry@gmail.com (N.K.S.); rtsharma312@gmail.com (R.S.); richasharmachem1@gmail.com (R.S.); mjain.chy@mnit.ac.in (M.J.); 2Department of Chemistry, Government Engineering College, Bharatpur 321303, India; 3Department of Medicinal Chemistry, National Institute of Pharmaceutical Education and Research, Raebareli (NIPER-R), New Transit Campus, Bijnor-Sisendi Road, Sarojini Nagar, Near CRPF Base Camp, Lucknow 226002, India; kshirsagarprasad06@gmail.com; 4Parasitology Laboratory, Department of Zoology, Panjab University, Chandigarh 160014, India; jyotijoshi86@gmail.com (J.J.); puzoology@yahoo.com (S.K.); 5Department of Chemistry, University of Delhi, North Campus, Delhi 110007, India; kunp93@gmail.com (K.P.); ramendrapratap@gmail.com (R.P.); 6School of Pharmacy, Macau University of Science and Technology, Avenida Wai Long, Taipa, Macau, China; 1909853vpa11003@student.must.edu.mo; 7Institute of New Drug Development, College of Medicine, China Medical University, No. 91, Hsueh-Shih Road, Taichung 40402, Taiwan

**Keywords:** microwave-assisted synthesis, spirooxindole, antileishmanial agents, molecular docking studies, structure–activity relationship

## Abstract

Leishmaniasis is a neglected tropical disease, and there is an emerging need for the development of effective drugs to treat it. To identify novel compounds with antileishmanial properties, a novel series of functionalized spiro[indoline-3,2′-pyrrolidin]-2-one/spiro[indoline-3,3′-pyrrolizin]-2-one **23a**–**f**, **24a**–**f**, and **25a**–**g** were prepared from natural-product-inspired pharmaceutically privileged bioactive sub-structures, i.e., isatins **20a**–**h**, various substituted chalcones **21a**–**f**, and **22a**–**c** amino acids, via 1,3-dipolar cycloaddition reactions in MeOH at 80 °C using a microwave-assisted approach. Compared to traditional methods, microwave-assisted synthesis produces higher yields and better quality, and it takes less time. We report here the in vitro antileishmanial activity against *Leishmania donovani* and SAR studies. The analogues **24a**, **24e**, **24f**, and **25d** were found to be the most active compounds of the series and showed IC_50_ values of 2.43 µM, 0.96 µM, 1.62 µM, and 3.55 µM, respectively, compared to the standard reference drug Amphotericin B (IC_50_ = 0.060 µM). All compounds were assessed for Leishmania DNA topoisomerase type IB inhibition activity using the standard drug Camptothecin, and **24a**, **24e**, **24f**, and **25d** showed potential results. In order to further validate the experimental results and gain a deeper understanding of the binding manner of such compounds, molecular docking studies were also performed. The stereochemistry of the novel functionalized spirooxindole derivatives was confirmed by single-crystal X-ray crystallography studies.

## 1. Introduction

According to the World Health Organization (WHO), in 2021, leishmaniasis emerged as an endemic in 99 countries/territories (out of 200 countries/territories), mainly in 4 eco-epidemiological provinces worldwide (the Americas, East Africa, North Africa, and West and South East Asia) [1,2]. It is caused by Leishmania, a protozoan parasite from the Trypanosomatidae family, which is transmitted by vectors and causes cutaneous leishmaniasis (CL), mucocutaneous leishmaniasis (MCL), and visceral leishmaniasis (VL); these are characterized by skin ulcers affecting the mouth, nose, and throat and “kala-azar”, respectively [3]. Kala-azar (visceral leishmaniasis) is the fatal form of the disease and is triggered by *Leishmania donovani*, an intramacrophage protozoan parasite transmitted by the bite of infected female phlebotomine sandflies. This lethal disease affects millions of individuals living in tropical/subtropical regions worldwide [4]. Approximately, twenty-one protozoan parasite species of Leishmania are responsible for causing leishmaniasis, and this is linked to a variety of symptomatology ranging from minor skin lesions at the bite site to the deadly visceral forms. A few standard drugs are available for the treatment of this disease, such as pentavalent antimonials, amphotericin B, its liposomal encapsulation (lamb-liposomal amphotericin B), and miltefosine. Amphotericin B emerged as an alternative second-line treatment for visceral, mucocutaneous, and cutaneous leishmaniasis, especially in the case of human HIV coinfection after resistance was reported in antimonials. In Thailand, amphotericin B is the only effective drug available for the treatment of leishmaniasis [5]. According to published research, there is no current safe and effective treatment for leishmaniasis. The antileishmanial medicines used to treat leishmaniasis at present are accompanied by various kinds of side effects, toxicity, and drug resistance [6,7]. As per the WHO report, approximately 700,000 to 1,000,000 new cases are reported every year [8]. Therefore, there is an urgent need for the advent of effective medications against this neglected tropical disease (NTD).

The spirooxindole class of bio-heterocycles are identified as privileged molecules and construct the core structural unit in several naturally occurring alkaloids such as horsfiline **1** [9,10], coerulescine **2** [11,12], marcfortine B **3** [13], spirotryprostatin A **4** and B **5** [14,15], elacomine **6** [16], formosanine **7** [17], pteropodine **8** [18], alstonisine **9** [19], rychnophyilline **10** [20], strychnofoline **11** [21], spirobrassinin **12** [22], mitraphylline **13** [23], notoamide A **14** [24], etc. (Figure 1). Spirooxindoles are blended with a wide range of biological activities such as antimicrobial [25,26], antimigraine activity [27], antitumoral [28], anti-inflammatory [29], antihelmintic activity [30], antimycobacterial [31], acetyl-cholinesterase inhibitory activities [32,33], anticancer activities [34,35,36], anesthetic [37], HIV-1 N-NRT inhibitor [38], antileishmanial [39], etc.

It has been well-documented that several pharmacologically privileged molecules can be assembled into a single structurally complex molecule with more multi-faceted and enhanced biological activities that can target biological sites of interest in a specific manner to combat specific diseases [40,41]. Several biologically active alkaloidal classes of heterocycles have been reported in the literature and show promising antileishmanial activity in vitro, ex vivo, and in vivo [42,43,44,45]. Recent studies have revealed that several substituted spirooxindoles **15**–**19** [46,47,48,49] show promising antileishmanial activity against promastigotes and the amastigotes forms of *Leishmania* (L.) species either in vitro or in vivo when treated with pentamidine, amphotericin B, or miltefosine as one of the standard drugs. Therefore, in our endeavor to search for novel bio-heterocycles as antileishmanial agents, we designed Prototype A, i.e., functionalized spiro[indoline-3,2′-pyrrolidin]-2-one/spiro[indoline-3,3′-pyrrolizin]-2-one incorporating subunits of **15**–**19** (Figure 2), and we assessed in vitro antileishmanial activities against the promastigotes form of L. donovani, with the expectation that a new series of amino-acid-based spirooxindole derivatives would also show promising in vitro activity. So far, a literature review has revealed that there is no report available showing amino-acid-based spirooxindoles as antileishmanial agents.

Microwave-assisted organic synthesis (MAOS) is a non-conventional, eco-friendly source of energy in chemical synthesis that can perform the reaction in a shorter time with less energy and furnish the product in a greater yield with higher purities as compared to traditional synthetic processes [50,51,52,53,54,55]. This fascinating method has a wide range of applications in drug discovery evaluation and the pharmaceutical segment for chemical synthesis. It has established an ongoing position in analytical and organic laboratory praxis [56]. Multi-component reactions (MCRs) via 1,3-dipolar cycloaddition reactions have been considered the best potential way for the synthesis of a library of spirooxindole derivatives [57,58].

Herein, we report the microwave-assisted synthesis as well as in vitro antileishmanial activity and structure–activity relationship studies of a novel series of functionalized spiro[indoline-3,2′-pyrrolidin]-2-one/spiro[indoline-3,3′-pyrrolizin]-2-one **23a**–**f**, **24a**–**f**, and **25a**–**g** via 1,3-dipolar cycloaddition. This was achieved by the interaction of various isatins and amino acids with substituted chalcones in up to 98% yields in a highly regioselective and stereoselective manner. For the first time, all the compounds **23a**–**f**, **24a**–**f**, and **25a**–**g** were prepared via microwave-assisted methodology. The stereochemistry of the novel functionalized spiro[indoline-3,2′-pyrrolidin]-2-one/spiro[indoline-3,3′-pyrrolizin]-2-ones was confirmed by single-crystal X-ray crystallography studies of the bromo derivative, i.e., compound **23f**. To the best of our knowledge, functionalized spiro[indoline-3,2′-pyrrolidin]-2-one/spiro[indoline-3,3′-pyrrolizin]-2-one **23a**–**f**, **24a**–**f**, and **25a**–**g** were identified for the first time as promising antileishmanial agents. In this study, amphotericin B was used as the standard reference drug. We also report the validation of wet results via in silico molecular docking studies of active compounds **24a**, **24e**, **24f**, and **25d**.

## 2. Results

### 2.1. Synthesis

The 1,3-dipolar cycloaddition reaction of azomethine ylides is a versatile reaction and is well known for the assembly of numerous varieties of complex bioactive azaheterocyclic skeletons [59,60,61]. The azomethine ylide is also reported to serve as an important building block for the construction of several natural-product-inspired aza-heterocycles [62,63,64] and bioactive molecules [65].

We commenced our synthetic investigation by taking isatin **20a**, chalcone **21a**, and L-proline **22a** as starting materials for carrying out microwave-assisted synthesis of spirooxindole-pyrrolidine **23a**. Initially, the reaction was attempted under refluxing conditions. Therefore, the reaction was carried out by taking **20a** (1 equiv.), **21a** (1 equiv.), and **22a** (1 equiv.) in MeOH under refluxing conditions for 120 min. We were delighted to get the desired spiro compound **23a** in 86% yield (Table 1, entry 1). Then, we analysed the effect of the number of equivalents of the starting materials. Thus, the reaction was repeated in MeOH with **20a** (1 equiv.), **21a** (1.5 equiv.), and **22a** (1.5 equiv.) under refluxing conditions for 180 min, yielding **23a** in 96% yield (Table 1, entry 2). It was noticed that changing the number of equivalents led to an improvement in the yield of the reaction. In order to reduce the time to complete the reaction, the reaction was subjected exactly to the same conditions as mentioned in entry no. 2 and allowed to run for 120, 60, and 30 min, which produced **23a** in 89%, 67%, and 58% yields, respectively (Table 1, entries 3–5). Keeping the reaction exactly under the same conditions as mentioned in entry no. 2, the screening of different solvents (AcCN, ethylene glycol, H_2_O, and ethanol) did not show an incremental effect on the yield of the reaction (Table 1, entries 6–9).

It is well known that microwave irradiation has been used as a fundamental tool for constructing aza-heterocycles with interesting properties, either in homogeneous or heterogeneous liquid reaction systems [66]. Utilizing the dual potential of both microwave irradiation as well as the 1,3,-dipolar cycloaddition reaction strategy; equimolar amounts of **20a** (1 equiv.), **21a** (1.5 equiv.), and **22a** (1.5 equiv.) dissolved in MeOH were treated under microwave irradiation conditions at 80 °C for 1 and 3 min, which produced **23a** in 41% and 71% yield, respectively (Table 1, entries 10–11). Intriguingly, when the same reaction was subjected to 5 min under microwave conditions; **23a** was obtained in 98% yield (Table 1, entry 12). The reactions were further screened with different solvents (AcCN, ethylene glycol, and ethanol) utilizing the same conditions as mentioned in entry no. 12 with varying times and temperatures (Table 1, entries 13–21). However, none of the reactions produced better yields than those obtained in entry no. 12. Therefore, equimolar amounts of **20a** (1 equiv.), **21a** (1.5 equiv.), and **22a** (1.5 equiv.) dissolved in MeOH under microwave irradiation conditions at 80 °C for 5 min was found to be the best optimized reaction condition (Table 1, entry 12).

Substituted isatins **20a**–**h**, substituted chalcones **21a**–**f**, and various amino acids **22a**–**c** were subjected to microwave-assisted 1,3-dipolar cycloaddition reactions in MeOH at 80 °C for 5 min, which produced the desired chalcone-isatin-amino-acid-based spirooxindole compounds **23a**–**f**, **24a**–**f**, and **25a**–**g** in excellent yields (up to 98%) in a highly diastereoselective manner (Figure 1, Please see Appendix A). In this reaction, [3 + 2] cycloaddition of substituted chalcones occurred with in situ generated azomethine ylides from microwave-assisted decarboxylative condensation of substituted isatins and various secondary amino acids.

The physico-chemical data, such as melting point and yield, under conventional conditions as well as in microwave-assisted conditions for all the compounds (**23a**–**f**, **24a**–**f**, and **25a**–**g**) are shown in Table 2.

The structures of all the synthesized compounds were well characterized by FT-IR, optical rotation, ^1^H-NMR, ^13^C-NMR spectroscopy, and HRMS mass spectrometric analysis (Please see Appendix A). Finally, the stereochemistry of the four chiral centres of the cycloaddition reaction was unequivocally determined by single-crystal X-ray diffraction analysis of the cycloadduct **23f** (Figure 3, please see Appendix A). After screening over the series of other derivatives, we found that the **23f** prepared in one step and obtained as an off-white solid in 86% yield, which was further subjected to crystallization; we were able to isolate the **23f** in ~10–11% yield using a slow evaporation crystallization technique with DCM as a solvent at low temperature. After couple of weeks, we came up with the single-crystal X-ray structure of the **23f**, the raw data of **23f** were subjected to the solution using Olex2 [67], and the crystal was crystallized in a trigonal system in R-3 space group. Consequently, the three-dimensional representation of compound **23f** shows that the compound has four chiral carbons, with one carbon having an R-configuration and the other three having S-configurations. The crystal structure confirmed that the trans-geometry of chalcone and the regioselectivity were also well established as a result of the concerted reaction of chalcones with the ylides.

### 2.2. Single-Crystal X-ray

Furthermore, it was observed that the crystallized framework had a hexagonal architecture consisting of six molecules in a circular fashion around the disordered functionality that takes non-planar circular conformations with the presence of short contacts in the alternate configuration. The molecular arrangement of **23f**, its inside functionality in a large cavity, and its size are directly proportional to the distance between the carbon atoms at the opposite sides and the Van der Waals radius of the carbon atom present in the ring. In order to understand more about the intermolecular interactions of **23f**, a Hirshfeld surface analysis using Crystal Explorer 3.1 software suite was used [68]. The 3D representation of short intermolecular contact can be provided by de and di mapped on the Hirshfeld surface, which corresponds to exterior and interior distances, respectively. The d_norm_, shape index, and curvedness of **23f** roughly indicating the presence of strong intermolecular short contacts and stronger Van der Waals interactions. The 2D finger plot of **23f** reveals significant interactions corresponding to C-C, H-C, and H-H contributes 4.5%, 8.9%, and 39.0% of the total Hirshfeld surface, respectively, which is again attributable to the presence of strong intermolecular interactions; these are more prominent that of π–π interactions (Figure 4).

As can be seen from Figure 1 and Figure 4, all synthesized spirooxindoles **23a**–**f**, **24a**–**f**, and **25a**–**g** were obtained in the 71–98% yield range. It was noticed that the reactions were occurring smoothly under microwave conditions with very good to excellent yields; however, the effects of the electron-donating group (EDG) and/or the electron-withdrawing group (EWG) had a marginal influence on the yield of the reaction. Among the spiro[indoline-3,2′-pyrrolidin]-2-one/spiro[indoline-3,3′-pyrrolizin]-2-one derivatives, i.e., **23a**–**f**, **24a**–**f**, and **25a**–**g**, it was noticeable that the EDG (Me, OMe, cycloalkyl, Cl, Br, and I), either on isatin or on Chalcone, produced the desired compound in excellent yield for **23a**, **23d**, **24d**, **24f**, and **25a**–**b**. However, in the case of EWG (NO_2_, F), either on isatin or on chalcone, the target compounds were obtained in a good-to-excellent yield range (Figure 5).

### 2.3. Biological Activity

Considering the importance of amphotericin B in the control of visceral leishmaniasis, the drug was selected as a control in the present study [69]. Camptothecin, a recognized inhibitor of LTopIB, effectively inhibits topoisomerase IB [70].Therefore, both drugs were used as control drugs for performing in vitro antileishmanial activity of all synthesized spirooxindoles **23a**–**f**, **24a**–**f**, and **25a**–**g**.

#### 2.3.1. In Vitro Antileishmanial Activity

The compounds (**23a**–**f**, **24a**–**f**, and **25a**–**g**) were initially screened for their in vitro antileishmanial activity against promastigotes of *Leishmania donovani* (MHOM/IN/1983/AG83) utilizing the Trypan blue dye exclusion method [71] and the plasmid relaxation assay using amphotericin B and camphothesin as standard reference drugs, respectively [72,73].

##### Trypan Blue Dye Exclusion Method

The promastigotes were harvested from the culture vials, counted, and 2 × 10^6^ cells/well were seeded in a 48-well culture plate. The antileishmanial screening of all the derivatives, **23a**–**f**, **24a**–**f**, and **25a**–**g**, as well as the positive control, was performed at various concentrations (2 μg/mL, 4 μg/mL, 8 μg/mL, and 16 μg/mL) added in triplicate. The plate was incubated at 22 ± 1 °C in the BOD incubator for 72 h. After 72 h, each well was counted for the number of viable parasites using the Trypan blue dye exclusion method, and the percentage growth inhibition was calculated by using the formula:$$\mathrm{Percentage}\;\mathrm{viability}=\frac{\mathrm{No}.\;\mathrm{of}\;\mathrm{viable}\;\mathrm{cells}\;\mathrm{in}\;\mathrm{treated}\;\mathrm{well}}{\mathrm{No}.\;\mathrm{of}\;\mathrm{viable}\;\mathrm{cells}\;\mathrm{in}\;\mathrm{blank}\;\mathrm{well}}\times 100$$
Percentage growth inhibition = 100 − percentage viability

The IC_50_ (inhibitory concentration at which 50% of the parasites were dead) value was obtained by plotting a linear dose–response curve in SPSS software (Version 23) [71].

##### Plasmid Relaxation Assay

The relaxation of supercoiled plasmid DNA is the method used for the determination of LTopIB activity. Various doses of each compound were treated with one unit of pure LTopIB (the enzyme to relax 0.5 µg of supercoiled DNA for 30 min at 37 °C) for 20 min at 4 °C. The reaction mixture, including 0.5 μg of supercoiled pBluescript SK(−) plasmid, 10 mM Tris-HCl buffer pH 7.5, 5 mM MgCl_2_, 0.1 mM EDTA, 15 µg/mL of bovine serum albumin, and 150 mM KCl, was then added in a final volume of 20 μL. After 30 min at 37 °C, the reaction mixtures were stopped by adding 4 μL of loading buffer, which included 5% sarkosyl, 0.12% bromophenol blue, and 25% glycerol. By electrophoresis, the topoisomers were separated on 1% agarose gels and electrophoresed at 2 V/cm for 16 h in a 0.1 M Tris-borate-EDTA buffer (pH 8.0) after being stained with ethidium bromide (0.5 µg/mL). Plotting the percentage of supercoiled DNA versus drug concentrations allowed researchers to determine the 50% inhibition concentration (IC_50_) values of LTopIB inhibition as the 50% reduction of supercoiled DNA [72].

#### 2.3.2. Inhibition of Leishmanial DNA Topoisomerase IB

Because of the presence of spirooxindole systems in the structure of these compounds, we aimed to assess their inhibitory potential on purified recombinant LTopIB measuring the relaxation of supercoiled plasmid DNA. In this regard, all spirooxindole derivatives were assessed for LTopIB inhibition through the prevention of DNA relaxation in a circular DNA plasmid. All compounds were tested at a single concentration of 100 µM to discard those that did not prevent DNA relaxation by LTopIB. After this initial test, potential inhibitor dose/response curves were performed to obtain their IC_50_ values. The compounds **24a**, **24e**, **24f**, and **25d** were potent LTopIB inhibitors, and the IC_50_ values of all nineteen compounds are shown in Table 3. The lowest IC_50_ value corresponded to **24e** (IC_50_ = 15.7 µM).

All nineteen compounds exhibited moderate-to-good antileishmanial activity against *Leishmania donovani*. The results are shown in Table 3.

#### 2.3.3. Structure–Activity Relationship (SAR) Studies

The inhibitory concentration (IC) values for all the spiro[indoline-3,2′-pyrrolidin]-2-one/spiro[indoline-3,3′-pyrrolizin]-2-one derivatives, i.e., **23a**–**f**, **24a**–**f**, and **25a**–**g**, and the positive-control drugs, were also determined against promastigotes of *Leishmania donovani* utilizing the Trypan blue dye exclusion method [68]. Amphotericin B was taken as a positive control. As can be seen from Table 3, compound **24e**, the most active compound of the series, showed potent in vitro antileishmanial activity, with the IC_50_ value of 0.96 µM against *Leishmania donovani*. Compound **24f**, the next most active compound in the series (IC_50_ = 1.62 µM), exhibited potent antileishmanial activity in comparison to the standard drug Amphotericin B (IC_50_ = 0.060 µM). Subsequently, compounds **24a** and **25d** also showed promising antileishmanial activity, with IC_50_ values of 2.43 µM and 3.55 µM, respectively. Furthermore, compounds **23d** and **24b** showed moderate activity (IC_50_ ≤ 10 µM). The rest of the compounds exhibited a lesser activity profile. Thus, SAR experiments indicated that the L-phenylalanine-based spirooxindoles showed a better antileishmanial activity profile as compared to L-proline and L-tryptophan-based counterparts. In proline-based spirooxindoles **23a**–**f**, the EDG group (OMe, Me) on the isatin moiety and the halogen (X = Br) on the chalcone functionality, i.e., **23d**, provided significant activity compared to Amphotericin B. Subsequently, phenylalanine-based spirooxindoles **24a**–**f** were found to be the best active compounds among the series. However, the presence of EDG (OMe, Me), EWG (NO_2_, F), or a halogen (X = Br, I) on the isatin moiety showed promising antileishmanial activity in compounds **24a**, **24b**, **24e**, and **24f** despite having EDG, EWG, or halogen groups on chalcone, except **24c**–**d**. Furthermore, among tryptophan-based spirooxindoles **25a**–**g**, the presence of EDG (OMe, Me), EWG (NO_2_, F), or a halogen (X = F, Br) on the isatin moiety showed moderate activity (**25d**) despite having EDG (OMe, Me, cyclohexyl), EWG (X = F), or a halogen (X = F, Cl, Br) on chalcone functionality. It was also observed that the presence of EDG or EWG on the spiroskeleton had no influence on the yield of the reaction.

### 2.4. Molecular Docking Studies

The molecular docking studies of the most active spiro[indoline-3,2′-pyrrolidin]-2-one/spiro[indoline-3,3′-pyrrolizin]-2-one derivatives, i.e., **24a**, **24e**, **24f**, and **25d**, were performed with *Leishmania donovani* topoisomerase I-vanadate-DNA complex protein (PDB ID: 2B9S) using Discovery Studio Visualizer Software [43]. 

#### 2.4.1. Ligand Preparation 

The two-dimensional structure (2d) of novel functionalized spiro[indoline-3,2′-pyrrolidin]-2-one/spiro[indoline-3,3′-pyrrolizin]-2-one-based compounds **23a**–**f**, **24a**–**f**, and **25a**–**g** along with standard drugs amphotericin and camptothecin were drawn in Chem Draw Ultra 22.0 software, and then the 2D structures of the ligands were converted into MDL molfile V3000 (*mol) format. The ligand was finally optimized with a small molecule protocol, which helps to remove tautomers, isomers, and duplicate conformations.

#### 2.4.2. Protein Preparation 

The protein crystal 3D structures of the heterodimeric L. Donovani topoisomerase I- vanadate DNA complex were taken from the protein data bank (PDB), PDB ID 2B9S. The protein was minimized using the simulation protocol via the CHARMm-based smart minimizer method, and protein preparation involved five different steps: cleaning protein, inserting missing atoms, refining loops, minimizing loops, and protonating protein. 

#### 2.4.3. In Silico Studies

Analysis of the docking results was carried out by comparing the binding affinities of all the proposed docked molecules to the complex protein. The docking of the abovementioned protein was carried out by removing DNA, and the remaining protein was kept in a grid box. Then, we explored the binding orientation of active functionalized spirooxindoles in terms of their Cdocker energy and Libdock score. It is to be noted that low Cdocker energy and high Libdock score values indicate higher binding affinity toward the target protein, thereby reflecting its higher potency (Table 4). 

The docking results for **24a** against *Leishmania donovani* showed a high binding affinity docking score indicated by a total score of 128.598, and it formed three H-bonds of length 2.19 Å, 2.9 Å, and 2.16 Å to the hydrophobic nucleophilic residues, i.e., the side chains of ASP: A-353 (aspartic acid), ARG: A-190 (arginine), and ASN: B-221 (asparagine), respectively. In the docking pose of the complex, the chemical nature of binding site residues within a radius of 3 Å showed non-bonding Van der Waals interactions with HIS: A-193 (histidine), ARG: A-314 (arginine), THR: B-217 (threonine), ILE: B-220 (isoleucine), and LYS: A-352 (lysine), thus leading to more stability and activity in this compound. In addition, 24a also exhibited a π–anion interaction with ASP: A-353 and an alkyl–π–alkyl interaction with the TYR: B-222 (thyrosine) amino acid residue (Figure 6).

The docking results for **24e** against *Leishmania donovani* showed a docking score of 96.0439 and showed attractive charges between nitro group substitution with ASP: A-353 (aspartic acid) and ARG: A-190 (arginine) of bond lengths of 4.58 Å and 3.80 Å, respectively, and LYS: A-352 (lysine) amino acids involved in H-bonds of lengths 2.37 Å and 2.23 Å with the carbonyl oxygen of the ligand. Furthermore, single carbon–hydrogen bond was observed with a bond length of 2.7 Å with ARG: A-190 (arginine), showing the presence of additional H-bonding (Figure 7).

Similar to compound **24e**, the docking profile for **24f** against the antileishmanial target showed a docking score of 131.125 and revealed non-bonding Van der Waals interactions with GLU A:182, LYS A:251 (lysine), ASN B:221 (asparagine), THR B:217 (threonine), and TYR B:222 (tyrosine). 

The π–anion interaction was observed between LYS A:352 (lysine) and isatin of bond length 3.85 Å. The conventional H-bonding of amino acids ASP: A-353 (aspartic acid) and ARG: A-190 (arginine) with bond lengths of 3.52 Å and 2.43 Å were present along with NH and carbonyl group moieties of ligand **24f**, respectively. Subsequently, **24f** also showed other interactions involving carbon–hydrogen bonds as well as π–cation, π–anion, and π–alkyl interactions (Figure 8).

The docking results for **25d** against *Leishmania donovani* (PDB ID: 2B9S) showed a high binding affinity docking score, indicated by a total score of 83.1911, and it mostly formed nonbonding Van der Waals interactions with amino acid residues GLY: A-189 (glycine), LYS: A-319 (lysine), HIS: A-193 (histidine), PHE: A-187 (phenylalanine), GLN: A-454 (glutamine), THR: B-217 (threonine), ARG: A-314 (arginine), ASN: B-221 (asparagine), ALA: A-324 (alanine), LYS: A-407 (lysine), and LYS: A-269 (lysine). It also shows a π–anion interaction between the phenyl moiety and ASP: A-353 (aspartic acid) and conventional hydrogen bonding between carbonyl oxygen and the amino acid residue, LYS: A-352 (lysine). The pair of alkyl–π–alkyl interactions HIS: A-453 (histidine) with a bond length of 5.24 Å and ARG: A-190 (arginine) with a bond length of 4.91 Å were also present, along with π–π T-shaped TYR: B-222 (thyrosine) with a bond length of 2.97 Å (Figure 9).

The docking results for camptothecin against *Leishmania donovani* showed a high binding affinity docking score indicated by a total score of 123.320 and formed two conventional H-bonds of length 2.55 Å and 1.79 Å to the hydrophobic nucleophilic residues, i.e., the side chains of ALA: A-324 (alanine) and LYS: A-319 (lysine), respectively. In the docking pose of the complex, the chemical nature of binding site residues showed non-bonding Van der Waals interactions with PHE: A-187 (phenylalanine), GLY: A-189 (glycine), THR: A-326 (threonine), SER: A-354 (serine), and GLU: A-353 (glutamic acid), which give extra stability and activity in this compound. In addition, camptothecin also exhibited the π–cation interaction with ARG: A-190 (arginine) and the alkyl–π–alkyl interaction with ALA: A-324 (alanine) and HIS: A-193 (histidine) amino acid residues (Figure 10).

## 3. Conclusions

In conclusion, we report the microwave-assisted synthesis of a novel series of functionalized spiro[indoline-3,2′-pyrrolidin]-2-one/spiro[indoline-3,3′-pyrrolizin]-2-one derivatives, i.e., **23a**–**f**, **24a**–**f**, and **25a**–**g**, and these have pharmaceutically privileged chalcones and amino acids. The time required for completion of reaction in MM varied from 5 min as compared to CM, which required 3 h. We also report, for the first time, the antileishmanial activity and SAR studies of **23a**–**f**, **24a**–**f**, and **25a**–**g**, which were validated by carrying out molecular docking studies of **24a**, **24e**, **24f**, and **25d**. The stereochemistry of the novel functionalized spiro[indoline-3,2′-pyrrolidin]-2-one/spiro[indoline-3,3′-pyrrolizin]-2-one derivatives were confirmed by single-crystal X-ray crystallography studies of **23f**. Among all the synthesized compounds, **24a** (IC_50_ = 2.43 µM), **24e** (IC_50_ = 0.96 µM), **24f** (IC_50_ = 1.62 µM), and **25d** (IC_50_ = 3.55 µM) showed potent in vitro antileishmanial activity against *Leishmania donovani* in comparison to the standard drug Amphotericin B (IC_50_ = 0.060 µM). All the compounds were tried as potential inhibitors of LTopIB, but only **24a**, **24e**, **24f**, and **25d** were able to inhibit the recombinant enzyme in vitro. Subsequently, the molecular docking studies validated the biological results. In short, our findings qualify the studied molecules as prospective antileishmanial agents with distinct pharmaceutically privileged structures that pave the way for further advanced applications.

## Data Availability

Not applicable.

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
