# Peer review of "Natural-Product-Inspired Microwave-Assisted Synthesis of Novel Spirooxindoles as Antileishmanial Agents: Synthesis, Stereochemical Assignment, Bioevaluation, SAR, and Molecular Docking Studies"

_molecules, 2023, doi:10.3390/molecules28124817_

Round 1
Reviewer 1 Report
Reviewer’s Comments to the Authors
The current research manuscript describes the Natural-Product-Inspired Microwave-Assisted Synthesis of Novel Spirooxindoles as Antileishmanial Agents: Synthesis, Stereochemical Assignment, Bio-evaluation, SAR, and Molecular Docking Studies. In the reviewer’s opinion, the manuscript has been organized appropriately. The reviewer feels that a potential reader can easily understand the details by consulting the main data outlined in the current version of the manuscript with few exceptions. The write-up is clear and to the point which is crucial for a research article with few exceptions. In the current version, the reviewer is left with the feeling that despite the significant efforts to conduct research work; this work cannot advance for publication in its present form. I would recommend publishing this work after minor modifications and addressing the raised concerns as follows:
1. The author tested only carbinol at 80 oC under MW conditions and got a 98% yield. However, ACN yielded 79% under reflux conditions but ACN was not tested under MW conditions. Similarly, ethylene glycol, were there particular reasons?
2. Why authors used wording like Carbinol? It would be much better to use MeOH.
- Although a plausible number of analogs with varying substitutions have been synthesized however the authors have detailed insufficient discussion on the SAR studies. Possible effects of Cl, F, Br MeO, Me, and other functional groups should also be provided in the SAR section.
- The compounds were chiral, but no optical rotation was determined in the current version.
- Table 2: The heading like spiro[indoline-3,2'-pyrrolidin]-2-one/spiro[indoline-3,3'- pyrrolizin]-2-one derivatives, should be changed.
Neither the structures of natural-product-inspired spiro-oxindole alkaloids from 1-14 nor the compounds 15, 16, 17, 18, and 19 (Figure 2) have -NO2 functional groups. But the analog 24a was most active against Leishmania donovani with a high binding affinity docking score indicated by a total score of 128.598. This is obvious due to the presence of the -NO2 group. The reviewer feels that most of the -NO2-containing analogs demonstrate failure in further drug discovery studies due to poor stability and ADMET studies. Why did the author select -NO2 substituted precursors?
Author Response
Dear Reviewer 1,
Attached please find the responses.
With Regards,
Sandeep Chaudhary

Reviewer 2 Report
The authors prepared series of functionalized spiro[indoline-3,2'-pyrrolidin]-2-one/spiro[indoline-3,3'-39 pyrrolizin]-2-one 23a-f, 24a-f, and 25a-g have been prepared from natural-product-inspired phar-40 maceutically privileged bioactive sub-structures, the following comments may improve manuscript quality:
1- The abstract needs editing regarding the main objective of the study and its impact on the research field as the authors clarify that in the introduction.
2- The references output needs revision and the introduction needs more references.
3- Figures resolution needs improvement (Figures 3, 5, 6, 7, and 8), in addition add A and B on the figures as the authors mentioned in the caption.
4- Add the titles of supplementary files on line 364.
Minor English editing is required.
Author Response
Dear Reviewer 2,
Attached please find the responses.
With Regards,
Sandeep Chaudhary

Reviewer 3 Report
Authors
The approach of your article is good; however, there are shortcomings in many aspects. I divide the comments into two parts.
Major corrections
1. In general, it is recommended to re-structure the order of the text, since some figures appear before being mentioned, and, in particular, in sections 2.1 and 2.1, ideas are mixed that must be well separated and specified. Each section must be well defined with its figures and that synthesis is not mixed with elucidation.
2. In the in vitro assay, amphotericin B is used as a control. It is necessary to support the reason for its use since its mechanism of action is as a membrane disruptor and its use is more related to an antifungal than an antiparasitic. It is necessary to argue or use as a positive control a drug (commercial use) or a molecule that has an inhibitory effect that coincides with the parasite and whose mechanism of action is well known and allows a better comparison to be established in vitro and that leads to its confirmation in silico analysis. IC50 results are shown in µg/mL when it is more appropriate for them to appear as either mM or µM concentration but not as expressed. It is necessary to make the suggested changes.
3. Methodologically, there are serious shortcomings in the molecular docking methodology and the SAR analysis, since there are no details of how the optimization of the ligands and the preparation of the protein were carried out. The data used for the geometric center of the protein, the size of the box used, and the exhaustiveness of the method, among other parameters, are not described in the docking.
4. Additionally, in the in silico assay, the topoisomerase 1 of L. donovani is used. However, the validation of the methodology by calculating RMSD is not shown, there is also no positive control for comparison and the use of this protein is not consistent with the mechanism of action of amphotericin, therefore the contribution of molecular docking is not comparable. The SAR analysis is very ambiguous and if you already have the IC50 data it is more convenient to do a QSAR.
5. Finally, there are 13 authors in the document, of which there are corresponding authors and two more participated equally. This number seems excessive to me, and their real participation should be reconsidered.
Minor corrections
6. Review the reference presentation format. Lines 60, 77, 111, to name a few.
7. Put the figures in the correct place, after being mentioned in the text. Modify tables in those cases where the information is repetitive, unifying cells (Examples Table 1 and Table 3).
8. Describe correctly in italics the descriptors R and S. Line 186.
9. In scheme 1, place the derivatives of the series 20 ai in the form of a list, for consistency with the rest of the scheme.
10. Line 208 writes correctly Van der Waals
11. Order the ideas and figures in sections 2.2 and 2.2 and avoid being repetitive in the information described.
12. Table 3 shows the IC50 values as the concentration in mM or µM.
I suggest a language review
Author Response
Dear Reviewer 3,
Attached please find the responses.
With Regards,
Sandeep Chaudhary

Round 2
Reviewer 3 Report
The manuscript proposal underwent the suggested modifications, and has an adequate distribution and information. However, for a work of this nature, I still think that the number of authors and the nature of their contribution is excessive. It would be very convenient if in future corrections present documents cleaner and better indicated as to the changes made; since a document with change control does not facilitate reading.
Minor editing of English language required. It is advisable to perform a review against plagiarism